# Targeting Stage-Specific Embryonic Antigen 4 (SSEA-4) in Triple Negative Breast Cancer by CAR T Cells Results in Unexpected on Target/off Tumor Toxicities in Mice

**DOI:** 10.3390/ijms24119184

**Published:** 2023-05-24

**Authors:** Rita Pfeifer, Wa’el Al Rawashdeh, Janina Brauner, Manuel Martinez-Osuna, Dominik Lock, Christoph Herbel, Dominik Eckardt, Mario Assenmacher, Andreas Bosio, Olaf T. Hardt, Ian C. D. Johnston

**Affiliations:** 1Miltenyi Biotec GmbH, 51429 Bergisch Gladbach, Germany; rita.pfeifer@miltenyibiotec.de (R.P.); janina.kuhl@miltenyibiotec.de (J.B.); manuel.martinezosuna@miltenyibiotec.de (M.M.-O.); dominik.lock@miltenyibiotec.de (D.L.); christoph.herbel@miltenyibiotec.de (C.H.); dominik.eckardt@miltenyibiotec.de (D.E.); mario.assenmacher@miltenyibiotec.de (M.A.); andreas.bosio@miltenyibiotec.de (A.B.);; 2Ossium Health Inc., Indianapolis, IN 46278, USA; wael.alrawashdeh@ossiumhealth.com

**Keywords:** CAR T cells, CAR design, on-target/off- tumor toxicity, SSEA-4, TNBC

## Abstract

Due to the paucity of targetable antigens, triple-negative breast cancer (TNBC) remains a challenging subtype of breast cancer to treat. In this study, we developed and evaluated a chimeric antigen receptor (CAR) T cell-based treatment modality for TNBC by targeting stage-specific embryonic antigen 4 (SSEA-4), a glycolipid whose overexpression in TNBC has been correlated with metastasis and chemoresistance. To delineate the optimal CAR configuration, a panel of SSEA-4-specific CARs containing alternative extracellular spacer domains was constructed. The different CAR constructs mediated antigen-specific T cell activation characterized by degranulation of T cells, secretion of inflammatory cytokines, and killing of SSEA-4-expressing target cells, but the extent of this activation differed depending on the length of the spacer region. Adoptive transfer of the CAR-engineered T cells into mice with subcutaneous TNBC xenografts mediated a limited antitumor effect but induced severe toxicity symptoms in the cohort receiving the most bioactive CAR variant. We found that progenitor cells in the lung and bone marrow express SSEA-4 and are likely co-targeted by the CAR T cells. Thus, this study has revealed serious adverse effects that raise safety concerns for SSEA-4-directed CAR therapies because of the risk of eliminating vital cells with stem cell properties.

## 1. Introduction

Being the most prevalent type of cancer among women worldwide, breast cancer causes more deaths than any other form of cancer in females [1,2]. Each year, more than 1.7 million new cases are diagnosed [3,4,5], of which approximately 15–20% develop lesions that are clinically described as “triple negative” due to their lack of expression of estrogen receptor (ER) and progesterone receptor (PR) as well as overexpression of HER2. The TNBC subtype differs from other breast cancers in its predilection to affect primarily women below the age of 40 [6], disproportionate occurrence of metastatic disease, and high incidence of cancer-related deaths [7,8]. Standard care chemotherapy with anthracyclines, taxanes, and/or platinum compounds is initially effective, but only 30–45% of patients achieve a pathological complete response and a comparable survival rate as other breast cancer subtypes [9,10,11]. When a pathological complete response is not achieved, the prognosis for TNBC patients is significantly worse than for other breast cancer subtypes [11]. This is largely because no oncogenic mutations are known that could be therapeutically targeted in a sizable fraction of triple-negative tumors.

More recently, the sialoglycolipid SSEA-4 was identified as a plasma membrane epitope of a TNBC subpopulation that is resistant to chemotherapy and displays high metastatic potential [12]. The glycolipid is part of the globoseries family and is composed of a hexameric carbohydrate structure that is covalently bound to ceramide [13]. SSEA-4 is widely used as a marker to identify pluripotent embryonic and mesenchymal stem cells, as its expression disappears quickly upon cell differentiation [14,15,16,17]. In oncology, SSEA-4 expression has been described in various solid tumors [18,19,20,21,22,23]. Intriguingly, a recent report linked SSEA-4 expression to the increased migratory potential of neoplastic cells, and an accumulation of the molecule was observed in protrusive membrane regions of metastatic cells, commonly known as filopodia and invadopodia [23]. As these structures contain metalloproteases that promote the degradation of the cellular matrix during cancer invasion and metastasis, it is fair to assume a role for SSEA-4 in malignant aggressiveness. In their work, Aloia and colleagues identified a subpopulation of ~10–30% of SSEA-4-positive cells in TNBC tumors, whose frequency increased by 2–3 fold following chemotherapy treatment [12]. Notably, the total number of SSEA-4-positive cells was unaffected by treatment; instead, these cells showed increased drug toxicity resistance, suggesting that SSEA-4 could serve as a marker for chemoresistant TNBC cells. Therefore, it was hypothesized that targeting SSEA-4 may provide a supportive therapeutic approach to TNBC that can be applied to patients after receiving chemotherapy and specifically target those tumor cells that escaped chemotherapy-induced cell death, thereby improving the therapeutic efficacy of TNBC treatment. As immune infiltration studies in breast cancers have shown TNBCs to have among the greatest incidences of patients with a robust immune infiltrate [24,25,26,27], a CAR T cell-based approach for therapeutic immunomodulation was selected. For this purpose, a set of second-generation chimeric receptors containing varying spacer regions within an otherwise identical CAR framework were constructed, and in vitro and in vivo analyses demonstrated functional differences among CAR T cells depending on the CAR architecture. A human/mouse cross-reactive single chain variable fragment (scFv) was incorporated into all receptors, enabling comprehensive profiling of the on-target/off-target tumor effects in the preclinical setting. This was possible due to the correlated expression of SSEA-4 that exists between human and mouse organisms. We show that the SSEA-4 CARs induced limited antitumor efficacy in vivo but, more importantly, could prompt severe toxicity in mice depending on the CAR architecture. The results obtained herein highlight the essentiality of careful target antigen selection in the translation of CAR T cell therapies to the solid tumor setting.

## 2. Results

### 2.1. Construction of SSEA-4-Directed CARs, T Cell Engineering, and Functional In Vitro Characterization

Numerous studies have shown that the spacer domain of a CAR needs to be tailored for each antigen epitope in order to maximize CAR T cell efficacy [28,29,30,31]. Hence, five SSEA-4-directed CARs incorporating either the IgG4 CH1-hinge-CH2CH3 (XL spacer; 326 aa), hinge-CH2CH3 (L spacer; 228 aa), IgG4 CH3 (M spacer; 119 aa), CD8α hinge (S spacer; 45 aa), or IgG4 hinge (XS spacer; 12 aa) region as a spacer domain were constructed with an otherwise identical antigen recognition domain from the human/mouse cross-reactive SSEA-4-specific mAb REA101, the CD8α-derived transmembrane domain, and the intracellular signaling domains of 4-1BB and CD3ζ. The marker gene ΔLNGFR was introduced downstream of the CAR gene to allow an analysis of transduction efficiency and enrichment of CAR-modified T cells. Upon initial expression analyses, which revealed inefficient surface expression of the M and XL spacer CAR on HEK293T (Appendix A) and T cells (Appendix A), the XS, S, and L spacer CAR constructs (Figure 1A) were selected for in-depth characterization. Transgene integration into primary T cells was achieved via lentiviral transduction, and the same standard conditions were used for each experiment to maximize reproducibility.

In order to consider variances in CAR expression efficiency, a congruent surface expression of the transduction marker ΔLNGFR was aimed for, both intra- and interexperimentally. By performing extended T cell transduction experiments with the different CAR candidates, it was consistently confirmed that each transfer vector displayed similar transduction efficiencies through ΔLNGFR-specific staining (Figure 1B). The frequencies of transgenic cells typically fell within the range of 35% to 60% for XS and S spacer CAR encoding vectors, while for L spacer CAR-encoding vectors, they ranged from 20% to 55%.

In addition, CD4+ T cells exhibited a slightly higher transduction rate compared to the CD8+ population for all donors and all viral vectors, as indicated by surface ΔLNGFR (Figure 1B). Direct detection of CAR expression was achieved by anti-mouse IgG (Fab-specific) or Protein L staining before and after ΔLNGFR enrichment. Further, the observation was made that anti-mouse IgG (Fab-specific) and Protein L staining approaches lead to different, non-congruent detections of CAR expression (Figure 1C). 

Having confirmed the surface expression of the synthetic receptors, we next carried out functional in vitro analyses using two breast cancer cell lines with varied SSEA-4 expression: the highly aggressive MDA-MB-231 TNBC cell line with a high density of SSEA-4 expression and the less-aggressive luminal A breast cancer cell line MCF-7 with a lower antigen expression (Figure 2A). When exposed to these cells, SSEA-4-directed CAR T cells degranulated and expressed surface CD107a, different from untransduced T cells (Figure 2B), demonstrating that the genetically modified T cells acquired SSEA-4-specific cytotoxic activity upon antigen engagement. Of note, although MDA-MB-231 and MCF-7 displayed varied SSEA-4 expression, the frequency of T cells incorporating the same CAR variant degranulated at comparable levels, suggesting that both antigen levels were sufficient to trigger CAR activity. In terms of T cell activation, both the S and L spacer CARs exhibited a trend towards greater potency compared to the XS spacer CAR, although the difference was not significant. Specifically, the XS spacer CAR managed to activate only approximately 20% of the transgenic CD8+ T cells, while the L and S spacer CARs prompted degranulation in over 40% or 50% of the cells. 

Furthermore, to then assess the kinetics of the cytolytic activity of CAR T cells by means of target cell death, dynamic monitoring of CAR T cell reactivity was performed using an automated imaging system (IncuCyte S3). While all groups of CAR T cells exhibited cytotoxicity against tumor cells, the various SSEA-4-directed CARs induced distinct kinetics of tumor elimination, revealing the following functional hierarchy, with the S spacer being the most reactive receptor: S spacer CAR > L spacer CAR > XS spacer CAR (Figure 2D). A similar efficacy pattern was evident when CAR T cells were analyzed for secretion of IFNγ, IL-2, and TNFα following antigen stimulation (Figure 2C). In summary, while all 3 chimeric receptors were able to induce effector function in T cells upon SSEA-4 stimulation, differences in the extent of T cell activation could be observed depending on the specific CAR spacer utilized. Furthermore, it was noticeable that there was no direct correlation observed between receptor activity and CAR spacer length. Surprisingly, the S spacer exhibited the highest level of effector function, followed by the L spacer, and then the XS spacer. 

### 2.2. Targeting SSEA-4-Positive Tumors In Vivo

After having established the in vitro functionality of SSEA-4-directed CAR T cells, we next sought to assess their therapeutic efficacy in vivo using an MDA-MB-231 cell line-derived xenograft model (Figure 3A). As shown in Figure 3B, when XS or L spacer CAR T cell therapy was administered, no antitumor effect was observed, whereas a slight decrease in tumor growth could be seen compared to the control group when the mice were treated with S spacer CAR T cells. However, on day 16 following adoptive transfer, the S spacer CAR T cell-treated cohort displayed severe morbidity symptoms including ruffled fur, hunched body posture, gasping, lack of motility, and weight loss (Figure 3C) and had to be euthanized. Interestingly, the body weight development in the group treated with the S spacer CAR (Figure 3C) showed a negative correlation with the presence of human CAR T cells in peripheral blood (Appendix A), indicating that the transgenic cells were proliferating throughout the therapeutic intervention and likely contributed to the observed toxicities. By contrast, mice treated with XS and L spacer CAR T cells did not exhibit any decrease in body weight (Figure 3C). Furthermore, a slightly increased number of CAR T cells was only seen in the peripheral blood of mice treated with L spacer CAR T cells, although their frequency was lower than that of the S spacer CAR-expressing T cells (Appendix A). Collectively, in vivo targeting of SSEA-4 by CAR T cells has been shown to mediate a weak antitumor response at best but was associated with adverse events in the context of S spacer CAR T cell targeting. Notably, the three different types of spacer CAR T cells followed the same reactivity trend in vivo as it was observed in vitro. Specifically, the S spacer CAR showed the highest bioactivity, followed by the L spacer CAR with intermediate activity and the XS spacer CAR with the lowest antigen reactivity, suggesting that the functional activity was mostly mediated independently of extrinsic factors; rather, the spacer length of the CAR had a profound effect on its bioactivity. 

### 2.3. Characterization of the In Vivo Toxicities Mediated by CAR T Cells

Immunohistochemistry of CD3 and ΔLNGFR on tumor sections revealed either a concentration of T cells at the invasive margin or absent lymphocyte infiltration at the tumor sites (Appendix A), suggesting that the observed toxicities were due to on-target/off-target tumor recognition. To confirm this, a high dose of 1 × 10^7^ XS, S, or L spacer CAR or untransduced control T cells per mouse was adoptively transferred into tumor-free mice. The high dose was chosen based on the observations in the clinical and preclinical setting that in cases of on-target/off-target tumor toxicities, high-dose administration of CAR T cells results in a rapid on-set of toxicities, and in order to avoid that potential GvHD effects interfere with delayed toxicity symptoms. Following systemic T cell administration, the mice were observed for any changes in behavior or physical health that could be linked to toxicity. Strikingly, symptoms were observed in all cohorts receiving SSEA-4-directed therapy, regardless of the type of CAR T cells used, although the severity of the symptoms varied depending on the specific CAR T cells. The S spacer CAR cohort displayed early signs of toxicity, including ruffled fur, hunched posture, tremor, and reduced signs of motility, mere hours after adoptive transfer. This was accompanied by a significant loss of body weight, averaging nearly 0.7 g per day. By day 6 after therapy started, the mice met the humane endpoint criteria (Figure 4A). By contrast, the L-spacer CAR-treated group exhibited toxicity symptoms of intermediate level, which emerged several days after adoptive transfer. By day 16, the group had to be taken out of the experiment due to humane endpoint criteria (Figure 4A). Although the XS spacer CAR T cell treatment did not exhibit obvious toxicity symptoms such as changes in body posture or fur such as the S- and L-spacer-treated groups, the humane endpoint was still met by day 16 (Figure 4A). 

Apart from the physical symptoms, toxicity was further evidenced by an elevation of serum levels of human IFNγ in all mouse cohorts receiving SSEA-4-directed CAR T cell therapy. Furthermore, the S spacer CAR T cell-treated group showed an increase in the levels of human IL-2 and TNFα in peripheral blood, as demonstrated in Figure 4B. Taken together, these findings suggested that targeting SSEA-4 in vivo resulted in significant on-target/off-target tumor toxicities, whereby the S spacer CAR T cell treatment conferred the highest degree of toxicity.

In order to investigate the pathology elicited from SSEA-4-directed CAR T cell therapy, an examination of the internal organs in mice treated with S spacer CAR and untransduced control T cells was conducted by a pathologist who was unaware of the treatment conditions. No evident differences in the structure of tissues and organs were observed between the two groups. Therefore, the organs were further analyzed by flow cytometry to determine the infiltration of CAR T cells, focusing on the blood, liver, lung, spleen, and bone marrow. Comparing the levels of ΔLNGFR expression, as indicated by median fluorescence intensity (MFI), the CAR T cells residing in the bone marrow and lungs exhibited higher expression levels than those found in human lymphocyte infiltrates in the blood, liver, and spleen (Figure 5A). It was hypothesized that gene integration favored genomic sites that are decondensed during T cell proliferation, considering the occurrence of transgene transfer in actively proliferating T cells that had undergone CD3/CD28-stimulated chromatin rearrangement. These sites are partially recondensed when T cells return to a steady state. However, antigen recognition-triggered proliferation of CAR T cells would reopen these sites, leading to enhanced transgene transcription.

To examine whether the lung and bone marrow served as active sites for CAR T cell proliferation, GFP-luciferase-expressing S spacer CAR or untransduced control T cells were generated using lentiviral engineering (achieving a transduction efficiency of 82%). Subsequently, 1 × 10^6^ CAR or untransduced T cells were intravenously injected into NSG mice without tumors (n = 2/group). The selected dose aimed to avoid injecting an excessive number of transgenic cells, as it would have resulted in a diffused luciferase signal present in the circulating CAR T cells, which would have made it challenging to identify the specific location of active CAR T cell proliferation. The luciferin signal was monitored regularly for a period of 3 days. Immediately after adoptive transfer, both CAR and control T cells migrated to the lungs, where they remained for several hours. However, as the observation time progressed, the control T cells started leaving the lungs and dispersing throughout the mouse body, resulting in a loss of localized luciferin signal. No evidence of proliferation was detected in these cells. In contrast, a fraction of lymphocytes carrying the S spacer CAR remained in the lung, and the luciferin signal continued to increase over time, indicating the active proliferation of these cells. Additionally, a distinct luciferin signal emerged in the bone marrow, demonstrating the highest growth rate (Figure 5B). Thus, it was concluded that the lung and bone marrow served as the primary sites for CAR T cell proliferation. Furthermore, an analysis of PD-1 expression by T cells infiltrating the bone marrow, lung, spleen, and liver (Figure 5C) confirmed that CAR T cell expansion was driven by antigen recognition. While minimal PD-1 expression was observed in untransduced T cells across all organs analyzed, CAR T cells significantly upregulated this activation marker in the lung and bone marrow. These findings suggested that the bone marrow and lung were the primary sites for on-target/off-target tumor recognition by SSEA-4-directed CAR T cells.

### 2.4. Identification of SSEA-4-Expressing Cells in Bone Marrow

A detailed examination of the bone marrow revealed that mice treated with 1 × 10^7^ SSEA-4-directed S spacer CAR T cells had a trend towards fewer live cells than those treated with untransduced control T cells (Figure 6A). Moreover, the therapeutic treatment induced a significant change in the cellular composition, as seen by an alteration in the ratio of murine CD45^+^ and CD45^−^ cells in the lineage-negative bone marrow compartment, as well as a strong decrease in the proportion of murine CD45+ cells (Figure 7B) and a loss of the CD45^+^Sca-1^bright^ population (Figure 6C). Given the lack of expression of SSEA-4 in lineage-positive bone marrow cells and its relative scarcity of 0.03% in the lineage-negative compartment (Figure 6D), it was hypothesized that the molecule is exclusively presented by hematopoietic progenitor cells. In line with this hypothesis, SSEA-4^+^ cells were found to co-express CD117 and Sca-1, which are both markers for undifferentiated cells (Figure 6E). Furthermore, the SSEA-4^+^Lin^-^Sca-1^+^CD117^+^ population expressed CD34, CD48, CD135, and CD150, which further confirmed that this subset was indeed multipotent progenitor cells (MPPs), as shown in Figure 6F. The decreased frequency of low-differentiated CD48^high^ cells following CAR therapy provided further proof that SSEA-4-directed CAR T cell treatment has an impact on MPP homeostasis. As CD48^high^ cells are the ultimate descendants of short-term hematopoietic stem cells and are generated by their self-renewal and differentiation processes (Figure 6G). The population of CD48^high^ cells within the Lin^-^CD45^+^Sca-1^+^CD117^+^ compartment was approximately 40–48% in mice treated with control T cells, whereas in mice receiving SSEA-4 CAR T cell therapy, this population was reduced to 27–33% (Figure 6G,H). Additionally, human CD34^+^CD38^+^ cells obtained from mobilized stem cell apheresis products of healthy donors were evaluated for SSEA-4 expression (Figure 6I). Around 30 out of 1 × 10^6^ cells were identified to be positive for the glycolipid, showing that, as in the murine system, SSEA-4 expression is also a characteristic of a subpopulation of blood progenitor cells.

### 2.5. Identification of SSEA-4-Expressing Cells in the Lung

Recently, hematopoietic progenitor cells with the ability to exhibit hematopoietic activities were discovered in the lung [32]. Consequently, the hypothesis was developed that the population targeted by SSEA-4-directed CAR T cells in the lung may include MPPs. To investigate this possibility, single cell suspensions were prepared from the lungs of untreated NSG mice and analyzed for SSEA-4 expression on both CD45^+^ and CD45^−^ populations. Despite the low overall prevalence of SSEA-4-expressing cells, which ranged between 0.3–0.6%, the major signal was surprisingly observed in the CD45^−^ compartment (Figure 7A), suggesting that the main target population in the lung was not of hematopoietic lineage. To characterize the SSEA-expressing cells, a multi-dimensional cytometry analysis was performed by sequentially staining 38 antibodies on the same lung specimen. Structurally, SSEA-4-positive cells were identified to either surround bronchiolar structures or localize within the tunica intima of CD31-positive blood vessels (Figure 7B), suggesting that various cell populations express the glycolipid. Intriguingly, both subpopulations lacked expression of the hematological marker CD45 and the epithelial marker CD326 but showed partial co-expression of CD34, CD44, and CD119 and complete co-expression of CD29, CD49a, CD81, and CD146 (Figure 7B). Thus, in the lung, vascular progenitor cells and mesenchymal stem cells express SSEA-4.

## 3. Discussion

TNBC is a particularly concerning form of cancer due to its dismal prognosis and few available treatment options. Monoclonal antibodies, immunization, small chemical inhibitors, and adoptive TIL treatments are only a few of the therapeutic strategies against TNBC that have been investigated, with relatively low success rates so far [33]. As CAR T cell therapy has demonstrated great success in the treatment of terminally ill patients suffering from hematological malignancies, this study sought to investigate the potential of transferring CAR T cell therapy to the treatment of TNBC. To this end, SSEA-4 was selected as a potential target antigen.

To identify the optimal CAR architecture, a panel of CAR molecules was designed and tested. Consistent with previous reports [28,29,30,34], this study further suggests that the spacer domain can play a crucial role in modifying the effector function of transgenic T cells. However, to our surprise, we find that the effectiveness of a synthetic receptor is not solely determined by the spatial constraints between the CAR and the antigen; in our study, a CD8 hinge and not an IgG CH2CH3 spacer CAR were the most efficient in recognizing a membrane-proximal epitope. 

This finding holds importance since prior studies primarily focused on IgG-derived sequences [30,31,34] and did not include a comparison with spacers derived from alternative parent proteins. Therefore, our research illustrates that the bioactivity of receptors in CAR T cell-target cell interaction is influenced not only by structural and spatial elements but also by other factors that are not fully comprehended at present. It is plausible that factors such as CAR flexibility and surface stability may hold greater significance than previously believed. Our most striking observation was that, despite a very small number of antigen-positive cells in non-tumorous mice, marked toxicities were observed upon SSEA-4-directed CAR T cell treatment. Thorough analysis revealed the glycolipid to be expressed by normal cells with a low differentiation status. Specifically, in bone marrow, SSEA-4 was expressed by MPPs, while in lung tissue, EPCs and MSCs were identified as expressing the antigen. Subsequent analysis of human CD34^+^CD38^+^ cells revealed a low prevalence of SSEA-4-positive cells, with only approximately 30 out of every 1 × 10^6^ cells expressing the glycolipid. Despite this low frequency, the mere presence of SSEA-4-expressing hematopoietic progenitor cells can result in adverse events in humans, as was seen in mice in this study, should treatment modalities that target SSEA-4 be applied. Moreover, it is reasonable to assume that the observed reactivity in the bone marrow is not only due to the targeting of MPPs but is the result of a complex process that involves several cell types. In support of this, Gang and colleagues reported SSEA-4 expression on MSCs, a cell subset that plays a critical role in hematopoiesis [15]. Thus, cellular distortion of the bone marrow compartment may not only be attributed to MPP-directed toxicity but also to MSC depletion. 

The destruction of lung-resident EPCs and MSCs could further explain the broad spectrum of toxicities observed in mice upon CAR T cell treatment. In particular, the depletion of both cell types likely leads to the disruption of lung tissue infrastructure, resulting in the early onset of gasping symptoms and cachexia. Moreover, growing evidence suggests that besides bone marrow and the lung, SSEA-4-positive progenitor cells can be found in other organs such as the pancreas [35] and kidney [36]. Importantly, da Silva Meirelles and colleagues showed that MSCs can be found in almost all adult tissues [37]. Thus, it cannot be excluded that these organs were not targeted during SSEA-4-directed CAR T cell treatment. One aspect supporting this hypothesis is the severe tremor, lack of motility, and weight loss observed in CAR T-cell treated mice, which can be due to the toxicity of adipose and muscle tissue-derived MSCs. Consequently, further investigations are needed to understand the extent of CAR T cell reactivity in organs other than the lung and bone marrow. Given the importance of MSCs, EPCs, and MPPs, it is intriguing that a systemic anti-SSEA-4 monoclonal antibody (mAb) treatment in the glioblastoma setting showed antitumor efficacy without any reported in vivo toxicities in a prior preclinical study [22]. Taking into account that Lou and colleagues targeted the same epitope on SSEA-4 as this study (unpublished data), it is remarkable that adverse on-target/off-target tumor events were seen only in this work. 

It is possible that the different nature of the therapeutics used may explain these inconsistent observations. T cells are known to elicit more pronounced pro-inflammatory immune responses compared to antibodies, and it is possible that the onset of potential toxicities in mAb therapy was delayed and may have occurred later than the 31-day therapeutic efficacy analysis performed by Lou and colleagues. Additionally, CAR T cells possess an overall avidity towards their antigen that is several magnitudes higher than that of the cognate antibody, resulting in a lower sensitivity threshold [34,38,39,40] Therefore, while mAbs may selectively recognize cells that highly express SSEA-4, CAR T cells may also recognize cells expressing the antigen at levels that are challenging for antibodies to detect.

This study underlines once again that the great potency of CAR T cells comes with a caveat, namely the risk of potentially lethal on-target/off-target tumor toxicities. Thus, while the antitumor response and oncogenic function of the antigen of choice should not be ignored, tumor selectivity should be the primary factor when selecting suitable antigens for CAR T cell targeting. This is key to achieving an acceptable safety profile during treatment. In fact, the absence of truly tumor-specific antigens has previously prevented CAR T cell therapy from being widely used in the context of solid tumors. As a result, attention has turned to target antigens that are highly expressed in solid tumors but display only minimal expression in normal tissue. Several clinical trials have tested the efficacy of this approach, but the results have been largely disappointing, as the antitumor efficacy has been limited and significant on-target/off-target tumor toxicities have manifested, even when the target antigen was expressed at very low levels on healthy tissue [38,41,42,43]. Studies as well as our findings emphasize the crucial need to confirm in future studies whether there is sufficient knowledge about the expression pattern of a potential target antigen on healthy tissue, including the significance of subpopulations expressing this antigen.

Taking into consideration that it is impossible to rule out that a promising tumor target may be co-expressed on vital healthy tissue, safety mechanisms are becoming increasingly unavoidable in CAR therapy in order to minimize the amount of damage that can be caused to non-malignant tissue. Therefore, great effort is being made to develop various strategies to mitigate the autoreactivity of CAR T cells, such as split or dual CAR approaches or various suicide mechanisms [44]. Nevertheless, an evaluation system that accurately compares the different approaches in a therapeutically relevant model is lacking. In light of this, SSEA-4 may serve as a model antigen to evaluate the different safety mechanisms in vivo and understand their advantages and disadvantages. In particular, it provides a platform to compare the safety and effectiveness of the different mechanisms against each other and can help identify potential problems before they arise in the clinical setting. Thus, targeting SSEA-4 as a model antigen holds promise to quickly recognize and eliminate unreliable technologies so that the clinical trial process can be streamlined and the most successful treatments can be implemented sooner into therapeutic applications.

## 4. Materials and Methods

### 4.1. Cells and Culture Conditions 

HEK293T cells, MCF-7, and MDA-MB-231 were purchased from the American Type Culture Collection (Manassas, VA, USA) and cultured in DMEM (Biochrom, Nuaillé, France) supplemented with 2 mM glutamine (Lonza, Basel, Switzerland) and 10% FCS (Biochrom, Berlin, Germany). MCF-7/eGFP-ffluc and MDA-MB-231/eGFP-ffluc were generated in-house by lentiviral transduction and subcloning. To support the growth of MCF-7, 17β-estradiol (Sigma–Aldrich, St. Louis, MO, USA) was supplied to the culture medium at a final concentration of 10 nmol/L. Cell confluency ranged typically between 20–80% during the culture maintenance phase.

Buffy coats and leukaphereses were obtained from the university hospitals in Cologne and Dortmund. All primary cellular products were derived from healthy donors after informed consent and cultivated in TexMACS medium (Miltenyi Biotec, Bergisch Gladbach, Germany) following processing.

### 4.2. Construction of SSEA-4 CARs

The CAR genes of interest were constructed using a commercial gene synthesis service in conjunction with an optimization algorithm to adjust the sequences for human codon usage bias (ATUM, Newark, CA). The SSEA-4-specific receptors contained an scFv of the anti-SSEA-4 mAB REA101 with a (G_4_S)3 linker joining the V_L_ and the V_H_ domains. To enhance the transportation of the synthetic receptors to the plasma membrane, a signaling peptide derived from the murine κ light chain was introduced N-terminally to the CAR sequence. The spacer region downstream of the scFv consisted of either the domain of IgG4 CH1-hinge-CH2CH3 (extra-large (XL) spacer; 326 amino acids), hinge-CH2CH3 (large (L) spacer; 228 amino acids), IgG4 CH3 (medium (M) spacer; 119 amino acids), CD8α hinge (small (S) spacer; 45 amino acids), or IgG4 hinge (extra-small (XS) spacer; 12 amino acids). In order to prevent possible interactions between the XL and L spacer CARs and cells expressing FcR, the APEFLG sequence within the CH2 domain of IgG4 was substituted with APPVA derived from IgG2. Additionally, an N279Q mutation was introduced to eliminate N-glycosylation at this specific site [30]. The spacer region was then connected to the transmembrane domain of human CD8α, the intracellular domain of 4-1BB, and the CD3ζ signaling domain obtained from UniProt. The gene sequences for the SSEA-4-specific CARs were then combined with a Furin-P2A sequence to enable the co-expression of the truncated low affinity nerve growth factor receptor (ΔLNGFR). To allow for transgene expression, a PGK promoter situated upstream of the respective CAR gene was engineered (Figure 1A). 

### 4.3. Lentiviral Vector Production

A second-generation system was used to produce self-inactivating VSV-G pseudotyped lentiviral vectors for CAR transduction. One day prior to transfection, approximately 1.6 × 10^7^ adherent HEK293T cells were seeded in each T175 flask, aiming for a confluency of 70–90% on the following day. At 24 h after cell seeding, each flask was transfected with a total of 35 µg plasmid DNA, consisting of a plasmid encoding VSV-G (pMDG2), a plasmid encoding gag/pol (pCMVdR8.74), and a transfer vector plasmid incorporating the transgene of interest. The transfection was performed using MACSfectin reagent (Miltenyi Biotech) at a DNA:MACSfectin ratio of 1:2, following the manufacturer’s recommendations. After overnight incubation, sodium butyrate (Sigma–Aldrich) was added at a final concentration of 10 mM. At 48 h post-DNA introduction, the supernatant was collected and clarified by centrifugation at 300× *g* and 4 °C for 5min, followed by filtration through 0.45 µm-pore-sized PVDF filters. To concentrate the lentivirus, the clarified supernatant was centrifuged at 4 °C and 4000× *g* for 24 h, and the resulting lentiviral pellets were air-dried and subsequently resuspended in cold PBS at a 100-fold concentration. Lentiviral aliquots were stored at −80 °C.

### 4.4. T Cell Isolation, Transduction, and ΔLNGFR Enrichment

Pan-T cells were isolated from peripheral blood mononuclear cells (PBMCs) obtained from buffy coats or leukaphereses through low-density centrifugation using negative magnetic selection (Miltenyi Biotec). Subsequently, pan T cells were resuspended at a concentration of 1 × 10^6^ cells per mL of TexMACS supplemented with IL-2 (40 IU/mL; Miltenyi Biotec), and T cell activation was achieved by stimulation with T cell TransAct (Miltenyi Biotec) according to the manufacturer’s protocol. At 24 h after activation, lentiviral transduction was performed using a multiplicity of infection (MOI) of 1.5. Throughout the culture period, an effort was made to maintain the cell concentration at 1 × 10^6^ cells/mL by adding fresh TexMACS medium supplemented with IL-2 (40 IU/mL) in a 2–3 day interval until day 12 or until enrichment for ΔLNGFR-positive cells. Selection of ΔLNGFR-expressing cells was routinely conducted on day 6 following activation using positive magnetic sorting with LNGFR MicroBeads (Miltenyi Biotec). Afterward, cells were expanded in the presence of IL-2 (40 IU/mL) until downstream processing.

### 4.5. Flow Cytometry Analysis

Cell surface expression of CARs was detected directly using goat anti-mouse IgG (Fab-specific) staining (Sigma–Aldrich) or Protein L (GenScript, Piscataway, NJ, USA), followed by anti-Biotin mAb (Miltenyi Biotec), or indirectly by anti-LNGFR staining (Miltenyi Biotec). Target antigen expression was assessed by anti-SSEA-4 (Miltenyi Biotec), and T cell phenotype was evaluated using CD4, CD8, and PD-1 (all from Miltenyi Biotec). Following xenograft experiments, human T cells were identified by human anti-CD45 (Miltenyi Biotec). All samples from in vivo experiments were stained in the presence of mouse and human FcR blocking reagents (Miltenyi Biotec). For identification of SSEA-4-expressing murine cells, the following mouse-specific antibodies were used (all from Miltenyi Biotec): CD24, CD29, CD31, CD34, CD38, CD41, CD43 Glyco, CD44, CD45, CD48, CD49a, CD49f, CD68, CD71, CD81, CD98, CD102, CD106, CD105, CD115, CD119, CD135, CD138, CD140b, CD144, CD146, CD148, CD271, CD309, CD326, Frizzled 1, Embigin, ESAM, Gr-1, Prominin-1, RAMP2, and Sca-1. The following fluorochrome-matched isotypes were used at the same concentration as the corresponding test antibodies (all from Miltenyi Biotec): mouse IgG1κ, mouse IgG2aκ, mouse IgG2bκ, rat IgG2aκ, rat IgG2bκ, REA. Isotype control for hamster IgG1λ was acquired from BD Biosciences (San Jose, CA, USA). For all mAb-based experiments, cells were stained for 10–15 min at 4 °C in the dark and washed twice with CliniMACS buffer (Miltenyi Biotec) containing 0.5% BSA (Miltenyi Biotec). To demarcate live/dead cells, propidium iodide (PI) was added directly before sample acquisition on a MACSQuant Analyzer 10 using the MACSQuant auto-labeling mode. For phenotypic characterization of SSEA-4-expressing cells, an effort was made to acquire 1 × 10^6^ viable cells/sample. The MACSQuantify software (Miltenyi Biotec) was used to analyze the acquired flow cytometry data. The gating strategy for each flow cytometry experiment consisted of three main parameters: identifying target cells based on appropriate forward scatter (FSC) and side scatter (SSC) settings; excluding aggregated cells using FSC-A and FSC-H plots; and excluding dead cells by gating on PI-negative cells. Only the cells that passed through these pre-gating steps were further analyzed for the expression of the antigens of interest. 

### 4.6. CAR T Cell Functional Assays

All in vitro assays were conducted 12–14 days post-T-cell TransAct activation with transgenic T cell populations enriched via ΔLNGFR and in the TexMACS medium without any additives. To ensure equal frequency among the T cell populations, untransduced T cells were spiked in to achieve identical transduction efficiencies. Expanded, untransduced T cells were used as a control for allogeneic reactivity.

To quantitatively evaluate the degranulation capacity of CAR T cells, the surface trafficking of CD107a was assessed upon antigen stimulation. For this, 2 × 10^5^ CAR T cells were co-cultured with target cells at a ratio of 1:1, and VioBlue-labeled CD107a mAb was added upon co-culture start. After 1 h of incubation, the cultures were supplemented with 10 µg/mL of both GolgiStop and GolgiPlug (BD Biosciences) and the cultures were incubated for an additional 4 h. The co-incubation was then stopped, and the cells were stained for CD4, CD8, and ΔLNGFR.

To study the kinetics of CAR T cell-mediated cytotoxicity, time-lapsed monitoring using the IncuCyte S3 system (Essen BioScience, Ann Arbor, MI) was carried out. To do so, 15,000 GFP-expressing MDA-MB-231 or MCF-7 cells were allowed to adhere overnight to the bottom of flat-bottomed 96-well plates, and the assay was started by adding equivalent numbers of CAR T cells to the cultures. Controls included cultures of target cells only and target cells with transduced control T cells. Phase contrast and green fluorescence images were captured every two hours for a period of 3–6 days at 10× magnification. The manufacturer’s software was used for image analysis with the following settings for GFP: Top-hat (100 μm and 2.0 GCU) with edge sensitivity of -38 and filters of 2000 μm^2^ area, 70.0 mean intensity, and 2 × 10^5^ integrated intensity.

For cytokine secretion experiments, CAR T cells were co-incubated with target cells at a 1:2 ratio using 5 × 10^4^ LNGFR+ effector cells and a total volume of 200 µL. After 24 h of incubation, 50 µL of the supernatant was harvested and either analyzed directly using the MACSPlex Cytokine 12 kit for human analytes (Miltenyi Biotec) or kept at −20 °C until further processing. Flow cytometric measurements and subsequent data analysis were carried out automatically using the MACSQuant Express Mode for MACSPlex.

### 4.7. In Vivo Studies

All experimental procedures were in compliance with European and German guidelines for the care and use of laboratory animals and were approved by the ethical committee on animal care and use in Nordrhein-Westfalen (approval number: 84-02.04.2016.A177). To establish subcutaneous (s.c.) tumors from cancer cell lines, female NOD/scid/IL2rγ-/- (NSG) mice (Charles River, Ecully, France) of 8 to 10 weeks of age were injected subcutaneously with 10 × 10^6^ eGFP-ffluc tumor cells resuspended in 150 µL PBS in the flank. When tumors became palpable at approximately 14 days, mice were randomized into groups and subjected to therapeutic treatment. Adoptive transfer of (eGFP-ffluc) CAR or mock T cells that had been expanded in vitro for 12 days occurred via tail vein injection. For therapeutic studies, a dose of 2 × 10^6^ CAR T cells in 100 µL PBS per mouse was administered based on the practical guidelines for dose conversion between humans and animals described by Nair & Jacob (2016). Untransduced T cell control was adjusted to the total T cell number injected in the CAR T cell groups. Peripheral blood was collected by puncturing the *vena facialis* using a lancet.

To study the observed toxicity, doses were increased to 1 × 10^7^ CAR T cells/mouse to cause any toxicities to manifest quickly and to prevent the possibility that GvHD effects may conflict with delayed toxicity symptoms.

To delineate the active sites of CAR T cell proliferation in non-tumor-bearing mice, a dose of 1 × 10^6^ CAR T cells/mouse was administered. This dose was chosen in order to ensure that not too many transgenic cells were injected, as otherwise the luciferase signal would also be visible in the circulating CAR T cells, creating a highly diffused signal that would make it impossible to detect the location of active CAR T cell proliferation.

To track the presence and quantity of both tumor and T cells in vivo, bioluminescent imaging was applied. For this, the cells were genetically modified to express luciferase, and in vivo, detection occurred via intraperitoneal (i.p.) injections of 3 mg D-luciferin (Gold Biotechnology, St. Louis, MO, USA) in 100 µL PBS into mice. After a 10 min interval of anesthesia, mice were imaged using an IVIS Lumina III imaging system (Perkin Elmer, Walluf, Germany). For anesthesia, the following protocol was applied: inhalation of 1.5% (*v*/*v*) isoflurane in oxygen, which continued during the measurement procedures with 0.5% (*v*/*v*) introduced via a nose cone (flow rate: 1 L/min). Image acquisition was conducted in auto-exposure mode with binning 8, field of view E, and F-stop 1.2. Analysis was conducted using Living Image software (Perkin Elmer) by manually drawing regions of interest (ROIs) around the mice, and luciferase activity for each mouse was quantitatively analyzed as total flux (photon/second). 

### 4.8. Ex Vivo Organ Preparation

For organ harvest, mice were euthanized by cervical dislocation, rinsed with 80% ethanol, and the organs of interest were excised. Storage occurred in the MACS tissue storage solution (Miltenyi Biotec) until downstream processing. Bone marrow was extracted from the femurs and tibias of mice by cutting off the epiphyses of the bones and rinsing the inner fragments. Upon breaking up the tissue by pipetting, the cell suspension was filtered through a 70 µm pore size MACS SmartStrainer and subjected to red blood cell lysis using RBC lysis buffer (both from Miltenyi Biotec). Spleen cell suspensions were prepared by gently mashing the organs with the plunger end of a syringe. Following the transfer of the splenocyte suspensions through a 70 μm pore-size MACS SmartStrainer, the cells were depleted of red blood cells with 1× RBC lysis buffer. The liver and lung were mechanically disaggregated with scalpels in MACS tissue dissociation solution as recommended by the manufacturer. For the lung, 2.5 mL and for the liver, 5 mL of dissociation solution per organ were used. Upon manual disaggregation, automated tissue homogenization was started using the gentleMACS Octo Dissociator (Miltenyi Biotec) and selecting the appropriate program settings (program for lung: m_LDK_37; program for liver: m_LIDK_37). After dissociation, the homogenized tissues were passed through MACS SmartStrainers with a 70 μm pore size, and red blood cells were analyzed using 1× RBC lysis buffer. 

Upon washing with CliniMACS buffer supplemented with 0.5% BSA, the prepared single-cell suspensions were subjected to flow cytometric analysis. Cell number determination was performed either automatically via the MACSQuant Analyzer or manually using the Neubauer chamber. 

### 4.9. MACSima™ Imaging for Lung Tissue Analysis

Characterization of SSEA-4-expressing lung cells was performed using the MACSima™ Imaging Platform, which allows one single tissue specimen to be stained sequentially with multiple biomarkers. The experimental procedure involved the design of a panel of 38 antibodies (for details, see the flow cytometry section) and tissue preparation by fixating 7 µm lung slices with acetone according to standard protocols using silanized glassware. Each MACSima™ Imaging cycle consisted of a 10 min incubation step with the respective antibody, washing, data acquisition, and bleaching. After the final cycle, the data were processed, and the generated data sets were analyzed and visualized using MACS iQ View (Miltenyi Biotec).

### 4.10. Statistical Analyses

Unless stated otherwise, the error bars in all graphs indicate the standard deviation of the mean. A non-parametric Mann–Whitney U test using GraphPad Prism 7.03 was used for statistical comparisons between the two groups. To simplify the presentation of the statistical analysis in graphs displaying several data points, a pairwise significance matrix was created(Figure 8), in which each box represents a comparison between two groups, as shown in Figure 1. A white box is shown if the difference is significant between two groups, whereas a black box is shown in the case of insignificant differences [45]. Additionally, the asterisk displayed in a white box denotes the level of significance with *, *p* < 0.05; **, *p* < 0.01; ***, *p* < 0.001.

## Figures and Tables

**Figure 1 ijms-24-09184-f001:**
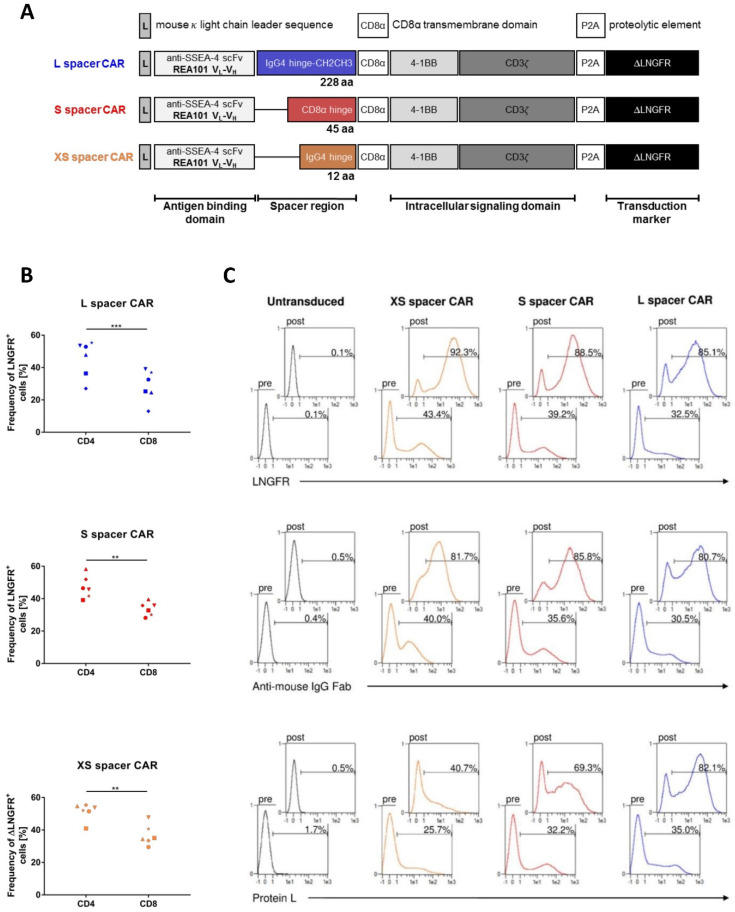
Design and expression of SSEA-4-directed CARs. (**A**) Schematic representation of SSEA-4-directed second-generation CAR variants used for functional evaluation. (**B**) Expression of ΔLNGFR on CD4+ and CD8+ T cells after the transduction of Pan T cells with XS, S, and L spacer CAR-encoding lentiviral vectors. Expression of ΔLNGFR was analyzed by flow cytometry 5 days after transgene transfer. The results are a summary of four independent experiments. A two-tailed paired *t*-test was used for statistical comparisons with **, *p* < 0.01, and ***, *p* < 0.001. (**C**) Flow cytometric detection of transgenic T cells before ΔLNGFR enrichment (pre) and after ΔLNGFR enrichment and expansion (post). Five days after transduction of Pan T cells with lentiviral vectors encoding the XS, S, or L spacer CAR variant, engineered T cells were enriched for ΔLNGFR by positive MACS selection and expanded for an additional 6 days. To detect the frequency of genetically modified T lymphocytes, cells were stained for ΔLNGFR using antigen-specific biotin-coupled mAb, while CAR expression was analyzed by either a biotinylated polyclonal goat anti-mouse IgG Fab-specific antibody or Protein L. Detection of the biotin conjugates was performed using α-biotin-APC secondary mAb labeling. Percentage of positive staining is indicated in each histogram. Results are representative of four independent experiments.

**Figure 2 ijms-24-09184-f002:**
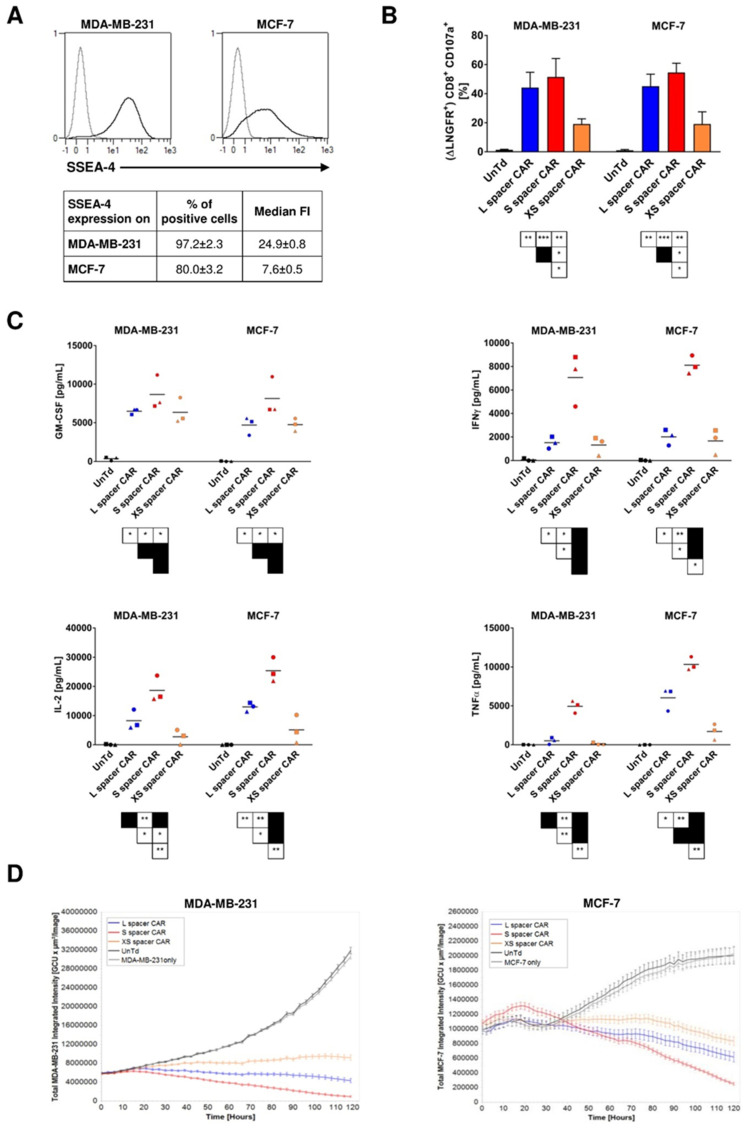
Functional profiling reveals a hierarchical activity of the different SSEA-4-directed CAR variants in vitro. (**A**) Surface expression of SSEA-4 on the breast cancer cell lines MCF-7 and MDA-MB-231. Antigen expression was determined by flow cytometry using SSEA-4 specific mAb REA101 (black line) and an isotype control (grey line). Histograms are representative for 3 independent experiments. The table depicts the averages of all 3 experiments. (**B**) Degranulation of CAR T cells after stimulation with MDA-MB-231 and MCF-7 cell lines. SSEA-4-specific CAR T cells and non-transduced control T cells were co-cultured with the indicated breast cancer cell lines at a ratio of 1:1 and in the presence of fluorochrome-conjugated CD107a mAb. After a 5 h incubation, CD107a externalization was measured by flow cytometry. Analysis was performed on the CD8+ population for non-modified control T cells and on the ΔLNGFR + CD8+ subset for transgenic T cells. Data represent the average of 4 donors and 3 similar independent experiments. (**C**) Cytokine secretion by CAR transduced T cells following antigen stimulation. On day 11 following transduction, CAR T cells were cocultured with the indicated cell lines at a ratio of 1:2 for 24 h and culture supernatants were analyzed for cytokine release using the MACSPlex technique. Culture of non-modified T cells with tumor cells served to assess the specificity of the CAR-mediated lymphocyte response. Results are a summary of 3 independently tested donors. (**D**) Dynamic monitoring of CAR T cell-mediated cytotoxicity. On day 11 after T cell transduction, MDA-MB-231 and MCF-7 cells stably expressing eGFP were co-cultured with different CAR T cell groups at a ratio of 1:1 and fluorescence emission was measured in the IncuCyte S3 imaging platform for 5 days every 2 h. Untransduced T cells and target cells alone served as the negative control. Shown is one representative experiment from 3 separate experiments and 5 donors in total. *, *p* < 0.05, **, *p* < 0.01 and ***, *p* < 0.001.

**Figure 3 ijms-24-09184-f003:**
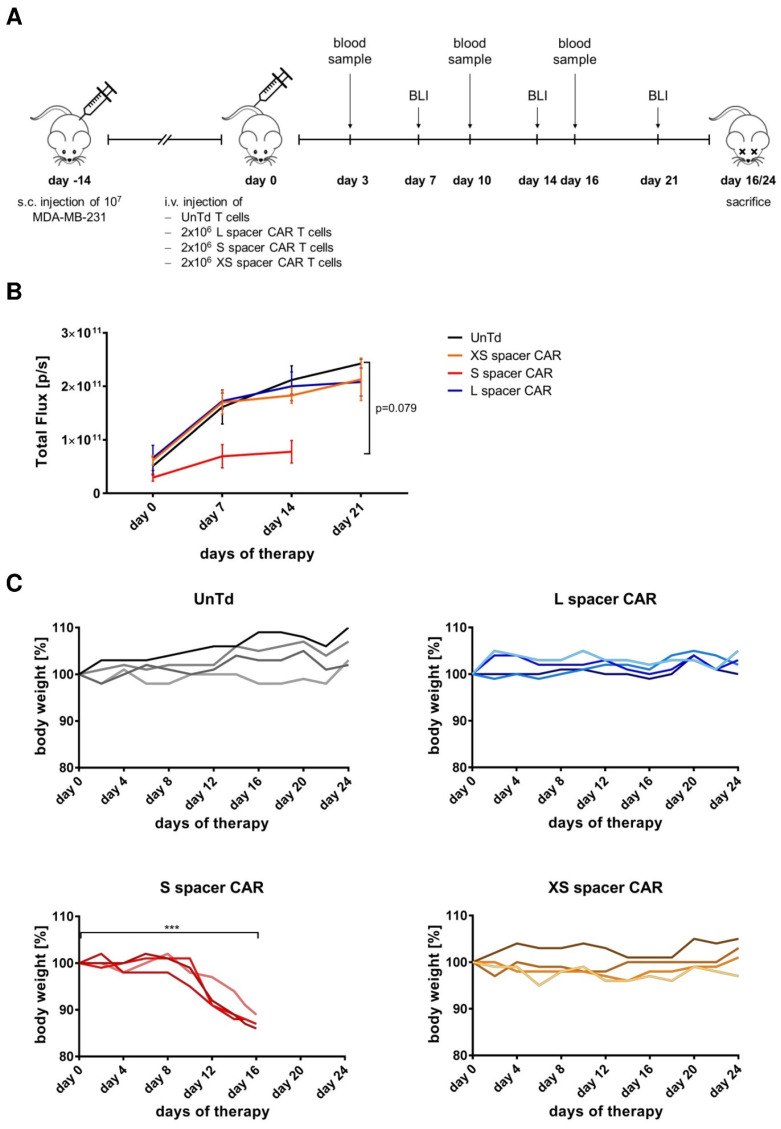
SSEA-4-directed CAR T cell treatments with different CAR variants confer limited antitumor activity, but can induce CAR construct-dependent toxicity in mice. (**A**) Outline of the experimental design to study the therapeutic potential of XS, S, and L spacer CAR T cells in a subcutaneous xenograft model. The study endpoint for each group was defined by the tumor reaching 1 cm in any direction, ≥20% weight loss, reduced motility or a combination thereof. For the S spacer CAR T cell-treated group, humane endpoint criteria were generally reached on day 16 and for the remaining cohorts on day 24. (**B**) Mean growth of s.c. MDA-MB-231 tumors in the 4 cohorts as determined by bioluminescence imaging (BLI) over time. Error bars represent SEM (n = 4/group). (**C**) Body weight development of mice treated with untransduced or XS, S, or L spacer CAR T cells throughout the therapeutic period. Results are a representation of two similar experiments and each line represents one mouse. A two-tailed paired *t*-test was used for statistical comparison of body weight at the beginning (day 0) and end of therapy start. ***, *p* < 0.001.

**Figure 4 ijms-24-09184-f004:**
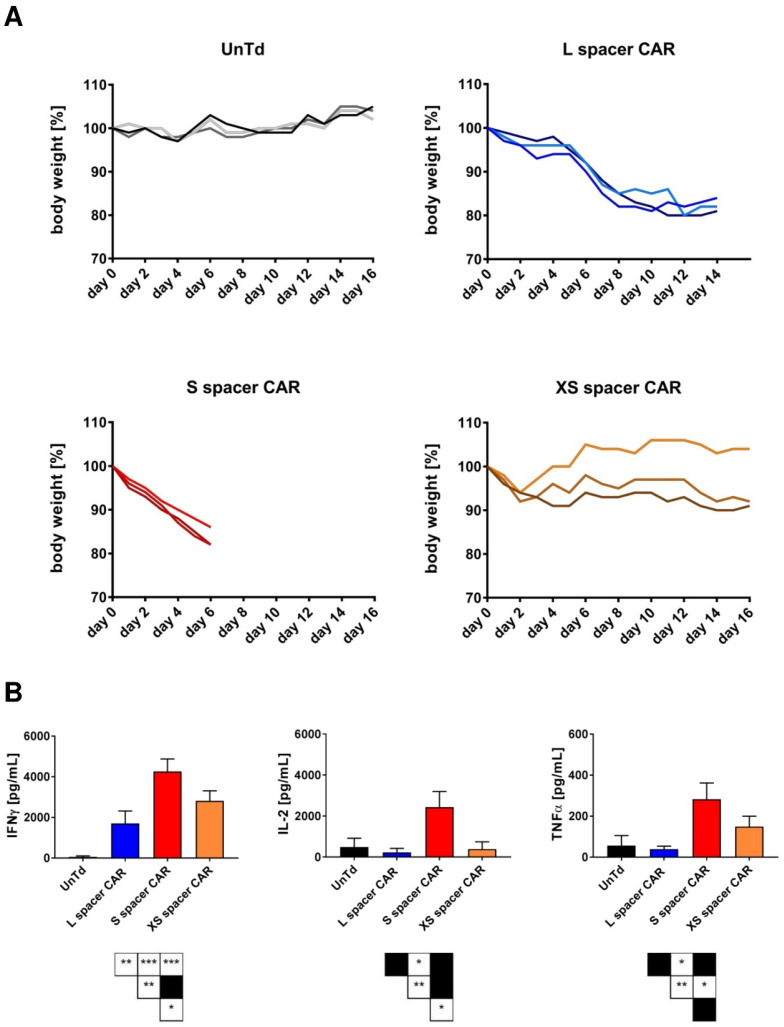
SSEA-4-directed CAR T cells induce varying levels of toxicity in tumor-free NSG mice. (**A**) Body weight development in naïve NSG mice following adoptive transfer of 1 × 10^7^ XS, S, or L spacer CAR T cells. For the injection of untransduced control T cells, the cell number was adjusted to the highest total T cell number injected in the CAR T cell groups. (**B**) Blood serum levels of human IFNγ, IL-2, and TNFα in non-tumor-bearing NSG mice 5 days after systemic CAR T cell treatment (n = 3/group). A non-parametric Mann–Whitney U test was used for statistical comparison, and no statistical significance was observed. *, *p* < 0.05; **, *p* < 0.01; ***, *p* < 0.001.

**Figure 5 ijms-24-09184-f005:**
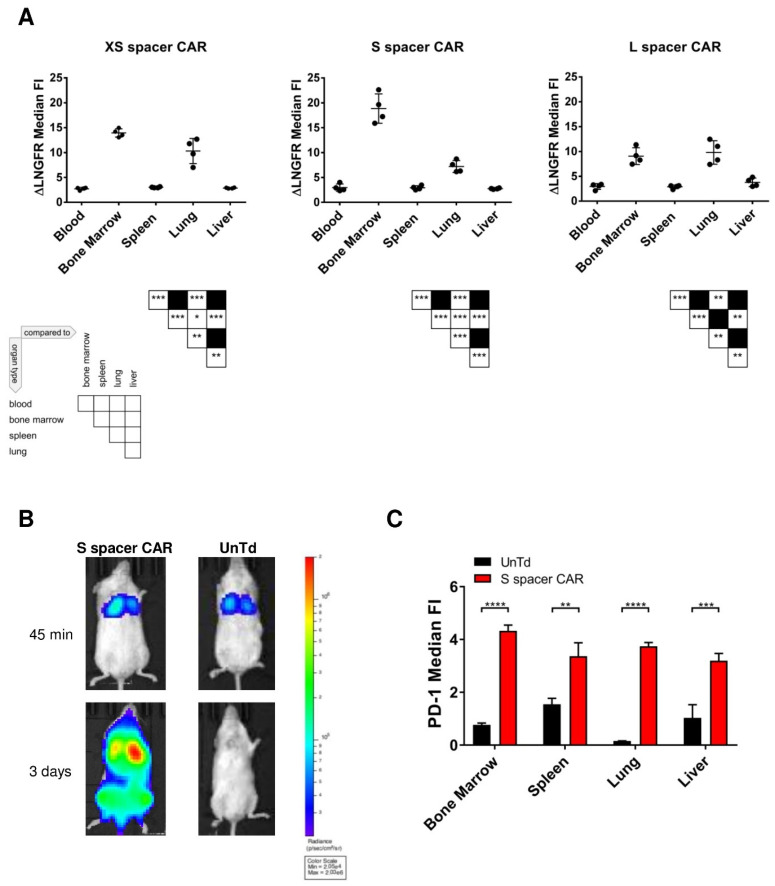
Following adoptive transfer into tumor-free NSG mice, SSEA-4-directed CAR T cells proliferate primarily in the lung and bone marrow. (**A**) Median Fluorescent Intensity (FI) of ΔLNGFR expression by gene-modified T cells infiltrating different organs. Sixteen (S spacer CAR) or twenty-four (XS and L spacer CAR) days after intravenous CAR T cell administration into NSG mice, blood, bone marrow, spleen, lung, and liver of the animals were harvested, dissociated into single cell suspensions, and ΔLNGFR expression was evaluated by flow cytometry. For analysis, gating was performed on viable human CD45+ single cells expressing ΔLNGFR (n = 4/group). (**B**) In vivo BLI of T cell trafficking and proliferation. Luciferase-expressing mock and CAR (S spacer) T cells were injected intravenously into NSG mice, and their biodistribution was analyzed at the indicated time points post-infusion (n = 2/group). (**C**) Median Fluorescent Intensity (FI) of PD-1 expression by untransduced and S-spacer CAR-transduced T cells infiltrating different organs. Three days after the adoptive transfer of T cells in NSG mice, bone marrow, spleen, lung, and liver were excised and processed into a single-cell suspension and PD-1 expression of the infiltrating human immune cells was analyzed by flow cytometry. For transduced cells, gating was performed on viable human CD45+ ΔLNGFR+ singlets; for untransduced cells, viable human CD45+ single cells were considered in the analysis (n = 4/group). A non-parametric Mann–Whitney U test was used with *, *p* < 0.05; **, *p* < 0.01; ***, *p* < 0.001; ****, *p* < 0.0001.

**Figure 6 ijms-24-09184-f006:**
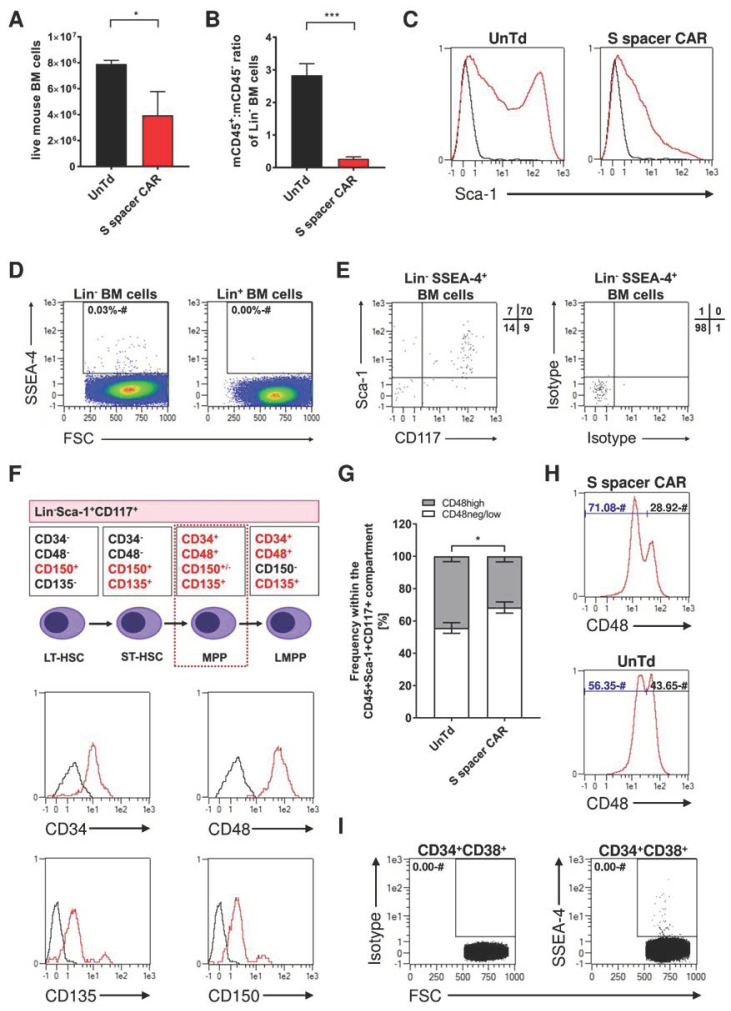
Bone marrow-resident SSEA-4-expressing cells show phenotypic markers of multipotent progenitor cells (MPP). (**A**) Total bone marrow cell counts following treatment with SSEA-4-directed CAR or untransduced control T cells. NSG mice were treated with 1 × 10^7^ S spacer CAR-expressing T cells, and 3 days later, the total bone marrow cell number was determined. Control group was treated with untransduced T cells, and the administered dose was adjusted to the total cell number used in the CAR T cell group. Data are the average number of cells isolated from the femurs and tibiae of one mouse (n = 5/group). (**B**) Mouse CD45+:mouse CD45- ratio of lineage-negative bone marrow cells after therapy. For analysis, lineage-positive mouse cells were excluded by Ter119-, CD5-, CD11b-, Gr-1-, Ly6B-, and CD45R-directed staining, and therapeutic T cells were excluded by human CD45-labeling (n = 5/group). (**C**) Sca-1-expression in the CD45 + Lin- bone marrow compartment after therapy. Viable single cells were preselected for analysis. The black line represents isotype control, and the red line represents Sca-1 staining. Histograms are representative of 5 samples per group. (**D**) SSEA-4-expression by lineage-negative and lineage-positive bone marrow cells of untreated NSG mice. Data are representative of three bone marrow samples analyzed. (**E**) Analysis of Sca-1 and CD117 expression by lineage-negative, SSEA-4-positive bone marrow cells. Numbers in quadrants next to the dot plot represent percentages of positive cells within the respective dot plot quadrant (n = 3/group). (**F**) Subcharacterization of SSEA-4-expressing Lin-Sca-1 + CD117+ (LSK) cells based on CD34, CD48, CD135, and CD150 expression. Isotype staining is represented by the black line and antigen staining by the red line. Data is representative of 3 stainings per group. (**G**) Proportion of CD48high versus CD48neg/low population within the LSK compartment after treatment (n = 4/group). (**H**) Representative histograms for CD48 expression by the LSK compartment after S spacer CAR or untransduced control T cell treatment. (**I**) Expression of SSEA-4 by human CD34 + CD38+ progenitor cells mobilized into the peripheral blood. Data is representative of two donors. A non-parametric Mann–Whitney U test was used for statistical comparisons with *, *p* < 0.05 and ***, *p* < 0.001.

**Figure 7 ijms-24-09184-f007:**
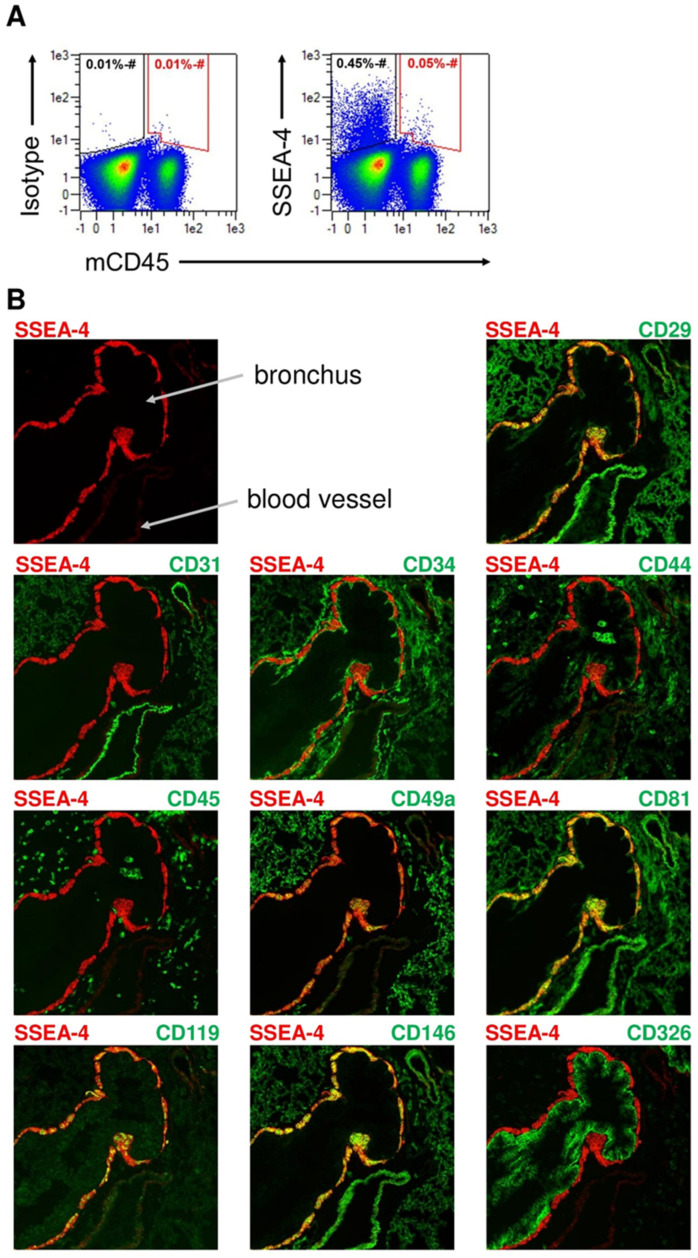
SSEA-4-positive cells in lung tissue co-express mesenchymal progenitor cell markers. (**A**) Expression of SSEA-4 on lung CD45+ and CD45- cells. Lungs of untreated NSG mice were dissociated, and the thus obtained single cell suspensions were subjected to flow cytometric analysis for correlated CD45 and SSEA-4 expression. An isotype control was used to assess background staining for SSEA-4. (**B**) Phenotyping of SSEA-4-expressing lung cells. Using multi-parametric cytometry analysis, a library of 38 antibodies was screened to identify cell surface markers that are co-expressed by SSEA-4-positive cells. A total of three lungs from untreated C57BL/6 mice were analyzed. Depicted are a selection of markers.

**Figure 8 ijms-24-09184-f008:**
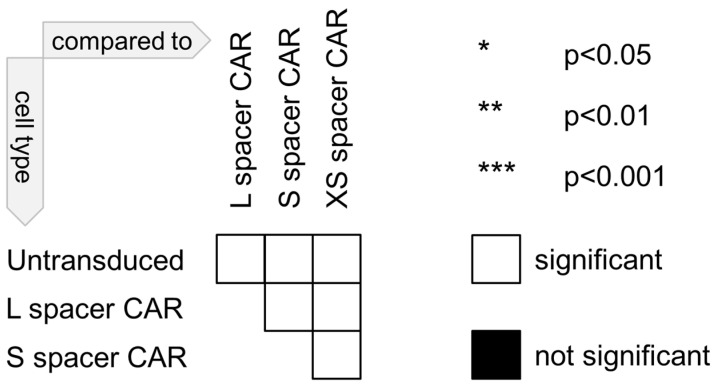
Organization of the pairwise significant matrix for group comparison.

## Data Availability

Not applicable.

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
