# Peer review of "Targeting Stage-Specific Embryonic Antigen 4 (SSEA-4) in Triple Negative Breast Cancer by CAR T Cells Results in Unexpected on Target/off Tumor Toxicities in Mice"

_ijms, 2023, doi:10.3390/ijms24119184_

Round 1

Reviewer 1 Report

This manuscript is very well written and has a logical progression throughout. The study highlights the potential toxicity issues associated with CAR-based therapies, that are perhaps not fully understood, and the challenges these present in CAR design. The study also reinforces the important role of, and learning from, protein design for CAR activity optimization.

The experimental design is appropriate and the authors presented a fair treatment of the data. This reviewer believes that this paper will be of considerable interest and should be published.

Author Response

We would like to thank the reviewer for taking the time and effort to review the manuscript and we value the positive feedback.

Reviewer 2 Report

In this manuscript, the authors present different constructs of CART-T cells targeting SSEA-4 and characterized their in vitro and in vivo effects. Conclusion was that only a minor antitumor activity was observed with the pore efficient construct. Moreover, this activity was accompanied by the emergence of a strong in vivo toxicity due to a cross-reactivity with normal mice cells expressing SSEA-4. Globally, the manuscript is clear and well structured. However, some part and especially the in vitro experiments could be better explained/commented/discussed.

In Discussion and Results, comments of in vitro experiments are poor. The last sentence describing the Figure 1  lines 109-110 should be changed. This summarizing sentence is useless, concluding on nothing. Is there any interesting data to extract from these experiments. If not,

Same remark about Figure 2 with the last sentence lines 143-145. One can notice activity of CAR-T are relatively similar on the 2 cell lines tested. Effects observed are systematically higher with the S spacer CAR, why that does not appear more clearly in the “summary” ?

One can notices in Figure 2A that population expressing of SSEA-4 is less homogeneous with MCF-7 than with the other cell line; And maybe there is  double peak, suggesting 2 populations. This point could be mentioned.

Since the S Spacer construct appeared more efficient in vitro, is not logical to observe with this construct stronger effect in vivo as well ? Why in vitro and in vivo experiments are nether related by the authors ?

How in vivo dose of CAR-T is defined ? Depending on the experiment 1 106, 2 106 or 1 107 cells were injected without any clear explanation. For construct without any activity the question of the dose could be discussed (even if it is probably not the major explanation).

Figure S4 should be modify with another sample of tumor treated by XS spacer CAR-T. In this sample, surprisingly there is far less antigen. Obviously, without SSEA-4, it is difficult to expect an activity of this construct. This picture raises the question of homogeneity of antigen expression in tumor cells injected to mice (the global SSEA-4 expression was only measured in vitro), a question not discussed in this manuscript.

Minor points:

Title of Figure 3 should be changed. Data presented in this figure show activity and toxicity only with one construct. Globalization is not possible here.

In Materials and Methods, the constructs of the CAR-T cells should refer to Fig 1A

No particular remark
